# On the Utility of Horizontal-to-Vertical Spectral Ratios of Ambient Noise in Joint Inversion with Rayleigh Wave Dispersion Curves for the Large-N Maupasacq Experiment

**DOI:** 10.3390/s21175946

**Published:** 2021-09-04

**Authors:** Maik Neukirch, Antonio García-Jerez, Antonio Villaseñor, Francisco Luzón, Jacques Brives, Laurent Stehly

**Affiliations:** 1Geosciences Barcelona, GEO3BCN-CSIC, C/Lluis Solé i Sabarís s/n, 08028 Barcelona, Spain; 2Department of Chemistry and Physics, University of Almería, Carretera de Sacramento s/n, La Cañada de San Urbano, 04120 Almería, Spain; agarcia-jerez@ual.es (A.G.-J.); fluzon@ual.es (F.L.); 3Institute of Marine Sciences, ICM-CSIC, Passeig Marítim de la Barceloneta 37-49, 08003 Barcelona, Spain; antonio.villasenor@csic.es; 4Institute of Earth Sciences (ISTerre), CNRS, University Grenoble Alps, University of Savoie Mont Blanc, and Gustave Eiffel University, 1381 Rue de la Piscine, 38610 Gieres, France; jacques.brives@univ-grenoble-alpes.fr (J.B.); laurent.stehly@univ-grenoble-alpes.fr (L.S.)

**Keywords:** horizontal-to-vertical spectral ratio, Rayleigh wave, velocity dispersion, joint inversion, shear wave velocity, Pyrenees, large-N experiment

## Abstract

Horizontal-to-Vertical Spectral Ratios (HVSR) and Rayleigh group velocity dispersion curves (DC) can be used to estimate the shallow S-wave velocity ($V_{S}$) structure. Knowing the $V_{S}$ structure is important for geophysical data interpretation either in order to better constrain data inversions for P-wave velocity ($V_{P}$) structures such as travel time tomography or full waveform inversions or to directly study the $V_{S}$ structure for geo-engineering purposes (e.g., ground motion prediction). The joint inversion of HVSR and dispersion data for 1D $V_{S}$ structure allows characterising the uppermost crust and near surface, where the HVSR data (0.03 to 10s) are most sensitive while the dispersion data (1 to 30s) constrain the deeper model which would, otherwise, add complexity to the HVSR data inversion and adversely affect its convergence. During a large-scale experiment, 197 three-component short-period stations, 41 broad band instruments and 190 geophones were continuously operated for 6 months (April to October 2017) covering an area of approximately $1500\;{\mathrm{km}}^{2}$ with a site spacing of approximately 1 to 3km. Joint inversion of HVSR and DC allowed estimating $V_{S}$ and, to some extent density, down to depths of around 1000m. Broadband and short period instruments performed statistically better than geophone nodes due to the latter’s gap in sensitivity between HVSR and DC. It may be possible to use HVSR data in a joint inversion with DC, increasing resolution for the shallower layers and/or alleviating the absence of short period DC data, which may be harder to obtain. By including HVSR to DC inversions, confidence improvements of two to three times for layers above 300m were achieved. Furthermore, HVSR/DC joint inversion may be useful to generate initial models for 3D tomographic inversions in large scale deployments. Lastly, the joint inversion of HVSR and DC data can be sensitive to density but this sensitivity is situational and depends strongly on the other inversion parameters, namely $V_{S}$ and $V_{P}$. Density estimates from a HVSR/DC joint inversion should be treated with care, while some subsurface structures may be sensitive, others are clearly not. Inclusion of gravity inversion to HVSR/DC joint inversion may be possible and prove useful.

## 1. Introduction

The horizontal-to-vertical spectral ratio (HVSR) or Nakamura’s technique [1,2,3] is widely used in seismic exploration for constraining shallow geologic structures [4]. While it is predominantly a tool for soft sediment thickness estimations and seismic hazard analysis [5,6], it draws increasing attention for permafrost depth estimation [7,8,9] and various subsurface characterisation purposes [10,11,12,13,14,15]. Among its advantages are practicality and cost efficiency [16], partly because, with very little additional effort, it is a by-product of most seismic survey campaigns that employ three component (3C) instruments. Therefore, the question arises as to whether or not principal seismic techniques, for which extensive large-N surveys are designed, can benefit from processing and analysing this essentially free data derivative even though the surveys’ targets may be beyond typical HVSR capabilities.

An exquisite laboratory to explore the answer to this question is the recent Maupasacq experiment, a dense network of 441 three-component seismic instruments placed on a rectangular grid in the French Pyrenees and dedicated to passive seismology research. The array covered about 1500 km^2^ with a minimum interstation distance of 1km and is surrounded by two rings of 24 broadband and short-period seismometers in total, located between 30km and 60km km from the center of the network, approximately. While the experiment attempts to shed light on the regional crustal structure of the Mauléon Basin in the Western Pyrenees with a large-N array, it is designed such that it allows inferring about the individual and joint performance of different passive seismic methods, such as local earthquake tomography (LET), ambient noise tomography (ANT) and, intrinsically, HVSR. Additionally, three different instrument types were employed for the experiment in order to establish the ideal use cases for each type. The survey’s acquisition layout was designed to maximise the information that could be obtained from LET and ANT with a target depth in the order of several kilometres [17]. These depths are beyond the capabilities of HVSR inversions in our experimental setup. Nevertheless, the inversion of dispersion curves experiences problems when a suboptimal initial range of near surface velocities causes low crustal velocities, and LET tends to overestimate near surface velocities, which may reduce overall resolution and accuracy. For both methods, HVSR can supply constrains on the near surface model parameters that allow distinguishing otherwise equally valid models and, therefore, complementing LET and ANT. Therefore, HVSR inversion results can potentially refine crustal velocities even though HVSR data cannot be used to sense such depths in our experiment. In fact, HVSR is already commonly and successfully used in joint inversions with dispersion curve (DC) data for shallow subsurface characterisations [18,19,20], where both methods effectively compensate for each others’ non uniqueness.

In this work, we illustrate that the semi-automatic inclusion of HVSR into the data inversion workflow reduces model variability for near surface $V_{S}$ and improves convergence to acceptable data fits. The study takes place within the framework of the Maupasacq experiment for which HVSR is a by-product of the data acquisition. First, we review the experiment design and the target area. Secondly, we present our methodology for semi-automatically inverting HVSR jointly with dispersion curve data for a large number of sites in order to obtain shallow model constraints. Then, we discuss data processing results and analyse statistically the 1D joint inversion results. Subsequently, we compare the results from 3D local earthquake tomography to our interpolated mean models. Lastly, by statistical analysis of data misfit and model variability, we estimate the benefit of including HVSR to 1D inversions of Rayleigh wave velocity dispersion for the shallow shear wave velocity ($V_{S}$) structure.

## 2. Maupasacq Experiment

Maupasacq abbreviates Mauléon passive seismic acquisition and the experiment constitutes a large scale passive seismic survey placed in the Mauléon Basin, SW France. The primary goal of the experiment is to image the survey area by applying various passive seismic methodologies, with each one contributing different perspectives and complementary information. In order to achieve this goal, an important part of the experiment is to explore the difficulties that entail campaigns with large numbers of stations for passive seismic acquisition and to find remedies where possible.

The Maupasacq experiment was mainly designed for the application of LET and ANT and a principal target depth at crustal scale, which is reflected by the choice of sensors, instrument spacing and array aperture (which will be discussed presently). Further details on Maupasacq are provided by [17]. However, a weak point of the two methods, reinforced by the design, is very limited near surface resolution. HVSR is sensitive to shallower structures; therefore, HVSR may help to evaluate and compensate some of the near surface resolution limitations of LET and ANT.

### 2.1. Study Area

The Mauléon–Arzacq rift system is situated in the French Basque country in the Western Pyrenees. The roughly EW-oriented system consists of four main domains (N to S): the Arzacq Basin, the Grand Rieu domain, the Mauléon Basin and the Axial domain [21,22]. In this study we focus on the Mauléon Basin bounded by the North Pyrenean Frontal Thrust (NPFT) to the north and the Igountze–Mendibelza Thrust (IMT) to the south. The Mauléon Basin is characterised by thick (up to several kilometres) Cretaceous sedimentary successions over hyper-extended crust. Coinciding with the centre of its Northern termination, the basin features a strong gravimetric anomaly (decreasing eastwards) that indicates elevated high-density lower crust or mantle rocks [22,23]. Figure 1 illustrates the regional geologic units and the dominant thrust systems and contextualises the experimental layout in Figure 2.

### 2.2. Acquisition

A dense network of 441 three-component instruments of three different types were installed over a period of 6 months (April to October 2017) covering effectively a total area of $1500\;{\mathrm{km}}^{2}$. The instrument pool consisted of 190 geophone nodes (SG-10 3C SERCEL), 197 short period instruments (3C Seismotech) and 54 broadband stations (Guralp CMG40, Trillium Compact and Trillium 120). The scope of the experiment is to image the crust with ANT and LET; therefore, the site layout has been optimised for these techniques. Site spacing is roughly a regular 3km grid for short period instruments and a 6–7 km grid for broad bands. Nodes are installed in five inline and three cross line configurations with a 1km site spacing along the grid as displayed in Figure 2. Ref. [17] analysed the collected data in detail and reported generally good data quality.

### 2.3. Purpose of Horizontal-to-Vertical Spectral Ratio

Data for HVSR are readily available due to the use of 3C sensors, and we can extract the available information and investigate its usefulness. Interpreting HVSR can provide insight to the structure of the very near surface at the scale of tens, at most hundreds of meters. However, the Maupasacq experiment focuses on crustal scale imaging techniques covering a large area; thus, that station spacing is, at best, 1km. This coverage can be considered fairly dense for an experiment on crustal scale, but such station spacing is hardly sufficient for accurately imaging the top 1km of a complex area.

HVSR measurements are usually acquired at much smaller interstation distances [15,24,25] so that adjacent stations yield similar and coherent data. The present site spacing is inadequate for the HVSR method (given the high operating frequencies of the instrumentation) but should be more dense in order to allow proper interpretation of the very near surface regional structure. Therefore, in this study, we focus on investigating the benefits of employing HVSR with DC inversions with respect to $V_{S}$ vertical resolution.

## 3. Data Processing and Inversion Methodologies

We focus on assessing the performance improvement due to HVSR for $V_{S}$ at the subsurface up to a depth of about 1km. We compare results directly between a 1D joint inversion of HVSR and DCs and (i) a pure DC inversion and (ii) independent ANT results.

Our methodology consists of three main steps. First, we obtain HVSR data from the Maupasacq data set. Second, we invert (a) DC and (b) the combination of HVSR and DC for 1D models of P− and S− wave velocity and density. In this step, the results of the 1D inversions are compared and evaluated statistically. Third, we compose a 3D model from all 1D models by smoothing and interpolation. The resulting 3D model is compared to the first 1km of the $V_{S}$ model obtained from LET [26,27].

### 3.1. Data Preparation and Processing

Rayleigh group velocity maps for the entire Maupasacq dataset have been obtained from ambient noise in [28] by analysing the correlation of the coda correlations between the measurements [29] We used the Rayleigh wave group velocity maps to infer the mean, confidence intervals and spectral covariance of the dispersion curve measurements at each site from the spatial variation around each site’s coordinates for a spectral range between 1s and 10s.

HVSR are computed for 196 short period, 41 broad band instruments and 190 geophones. Each sites’ mean, variance and covariance of the HVSR are obtained from the temporal variation of HVSR estimates based on successive intervals of 15min for a spectral range from 0.025s to 10s (0.5s for geophones due to a higher cut-off frequency) based on the Hilbert–Huang transform method presented by [30].

It has been ensured that the number of HVSR data samples equals the number of DC data samples at each site in order to avoid the need to rebalance weighting during the inversion and reduce the risk to bias the inversion towards one or the other data type.

### 3.2. Inversion of DCs and Joint Inversion with HVSR

The inversion of DC data and the joint inversion of HVSR and DC was carried out with the computer program described by [31] using the mean and covariance for $\chi^{2}$ misfit calculation. The inversions are carried out in five distinct steps with an increasing number of layers with fixed thickness and increasing parameter freedom. At the beginning, we define a model with nine layers of logarithmically increasing thickness, which we expand to 16 and 30 layers. In these first three steps, the parameters $V_{P}$ are free within reasonable bounds, $V_{S}$ is free but must always increase with depths, and density is set at a constant $2\;\mathrm{kg}/m^{3}$. In the following steps, first $V_{S}$ and then density becomes free parameters.

We refer to Table 1 for an overview of the constant layer thicknesses and to Table 2 for a summary of the steps. The five step inversion strategy is repeated 20 times with random starting models in order to ensure that the converged results are stable. All tested models in all steps and all repetitions are saved and result approximately in a total from 100,000 to 200,000 models at each site. The misfit between observed data and modelled data, $\chi^{2}$, is assumed to represent the negative log-likelihood of the evaluated 1D models.
(1)$$p=\exp\left(-0.5\chi^{2}\right).$$

All evaluated models’ mean and standard deviations are computed with weights according to the models’ estimated probability [30] (Equations (18) and (19)). Sites for which the lowest $\chi^{2}$ achieved are larger than 14 are discarded (we will provide more details in Section 4.2 and Section 4.3). Note that we only interpret layers above 1km below each site. All layers below 1km only serve (i) to improve convergence, (ii) to stabilise the 1D inversion and (iii) to reduce inversion artefacts in the upper layers; these layers are not interpreted further.

### 3.3. 3D Model Composition

The 3D models are constructed from interpolation of the mean 1D models. We use the mean models, instead of the best models, because they (1) likely contain fewer artefacts from over-fitting, (2) are generally more smooth and (3) can be easily co-interpreted with the computed standard deviation. Subsequently, the final 3D model is smoothed with a boxcar function to better represent the achievable resolution and further counter overfitting effects. The horizontal (vertical) size of the boxcar is 3km (100m) to ensure that the information in the 3D model represents typical regional structure that is supported by the measurements at several neighbouring sites.

## 4. Results

### 4.1. Horizontal-to-Vertical and Dispersion Curves

The obtained HVSR curves are complex and typically contain 2–3 peaks across the frequency range, of which usually one is dominant. Rarely do we find no peaks present or more than three. Although there is normally a major peak present, the HVSR data indicate a complex subsurface. Often the major peaks are not sharp but flattened, widened and asymmetric. Secondary smaller peaks further complicate interpretation. Poor confidence estimates allow for the presence of unidentified peaks too. Left hand panels of Figure 3 and Figure 4 illustrate examples of HVSR data.

The complexity of the subsurface becomes apparent when we plot the major peak frequencies as a map (see Figure 5) in an effort to identify regions with similar structure, e.g., depressions. The frequency peak map displays high complexity and confirms our suspicion that HVSR data with the available site spacing are hard to interpret, e.g., to find consistent regions. In order to ease interpretation, we smooth and interpolate the map and draw contours. Despite the smoother image on which we might be able to identify areas of interest, the peak frequency map of such a large area remains difficult to work with. The three complicating issues are the fact that there can be more than one interface causing the peak frequency, the inhomogeneous topography and the varying average $V_{S}$ above the main Vs contrast. The first issue is present because, in this study, our target depth is substantially deeper than the very near surface that is typical for HVSR studies, which only consider the first notable interface in $V_{S}$ structure and often estimates associated depth to the peak frequency by $d_{p}=\frac{V_{S}}{4f_{p}}$. However, typically the entire structure contributes to present HVSR responses, with decreasing relevance as depth increases; thus, deeper interfaces can still play important roles especially when multiple/flattened/widened peaks are present. Both topography and $V_{S}$ must be relatively constant for a direct interpretation of the peak frequency map, which is clearly not the case here. In order to interpret the obtained peak frequencies to some extent, we use the Rayleigh wave group velocity at 1 s (the shortest period at which it is available) as a proxy for shallow $V_{S}$ and correct for topography so that the associated depth $d_{p}=\mathrm{altitude}-\frac{V_{S}}{4f_{p}}$ yields a first approximation of the elevation (a.s.l.) of the main velocity contrast (Figure 5d). Figure 5 illustrates the peak frequencies for each of the sites, a smoothly interpolated peak frequency map, the interpolated maps of altitude, the mentioned proxy for the shallow $V_{S}$ [28,32] and the interpolated depth associated with the local velocity and peak frequency corrected for the altitude. In conclusion, we find that interpretation of peak frequencies and associated depth is daunting in this scenario and does not provide much insight, especially because it is clear from the HVSR curves that the subsurface must be much more complex than what can be derived from such a simplistic interpretation. Therefore, we must perform data inversion in order to identify subsurface structures from HVSR data more accurately, but we already note that the data from this study area appear to be borderline 1D at best, which might not meet the requirements of state-of-the-art HVSR inversion algorithms. Artefacts due to over-fitting and the inability to fit some data may be expected. While we can still learn from carefully interpreted 1D inversion results, a real 3D inversion would provide more reliable insight but is not yet technically available.

### 4.2. Benefit of 1D Joint Inversion of HVSR and DC

While it is known that HVSR and DC data complement each other, here we want to study what specific improvement can be expected from a joint inversion. For this, we compare results from an inversion of only DC data with the results obtained from the joint inversion. It can be expected that the data misfit is usually better for the inversion of a single data type, because a joint inversion must inherently find a compromise between all data so that the data misfit is not a suitable performance measure for this study. Instead, we compare the final model confidence, which reflects how well a model can be ascertained with a given inversion strategy. Final $V_{S}$ models’ variances are displayed in Figure 6 for DC inversions and joint inversions of DC and HVSR for data of all short period stations. The obtained histograms are displayed on a logarithmic axis for the variances and are fitted to a (log-)Gaussian distribution to obtain mean values and confidence estimates. At depths up to 300m, the HVSR/DC joint inversion improves $V_{S}$ variance estimates of each layer by a factor of 2to3 according to maximum error propagation. At depths larger than 300m, the layer $V_{S}$ variance estimates are equal between both inversion strategies. These results also indicate that the sensitivity of the present HVSR data reaches at least 300m before it declines, while the present DC data are most sensitive for depths larger than 300m. It is important to note that for an optimal joint inversion, both data types, HVSR and DC, must overlap sufficiently with their respective depth sensitivity as it is the case for the present data of short period sensors and broad band stations. Below, we will expose results for data of geophone sensors without sufficient sensitivity overlap and the corresponding issues that can be observed. The depth sensitivity of both methods depends on the available data bandwidth, which means that, in practice, adequate HVSR data can be used to partially amend the absence of short period DC data that may be more difficult to obtain.

### 4.3. 1D Joint Inversion

Most of the sites achieved a moderately good misfit with $\chi^{2}<14$ (see Figure 7a), with a better performance for short period and broad band sites. The misfit cut value of 14 was chosen manually by visual inspection of the data fit and considering that a significant amount of models can be interpreted for the better performing instrument types. The tested models are evaluated for the weighted mean and standard deviation of $V_{S}$ and density according to each models’ log-likelihood. Examples of the processed and modelled HVSR and dispersion curves are plotted in the left hand panels of Figure 3 and Figure 4. Dark shades correspond to the 99% confidence interval in the measured data and the light shades are associated with the estimated standard deviation of the models. In the right hand panels, red colours correspond to $V_{S}$ and blue colours to density, while the shade illustrates the standard deviation of the model parameters. Clearly, $V_{S}$ estimates are more precise, and sensitivity to density is often lower than our resolution, which can be observed when, for example, the constant value of $\rho =2tm^{-3}$ (the initial value) is within the estimated confidence interval.

Short period and broad band instruments result in very similar data misfit, model convergence and model variances, but geophone nodes exhibit poor convergence to an inferior data misfit with bi-modal model variances (Figure 7). The poorer performance of the geophone nodes is caused by the limited HVSR data bandwidth due to the relatively high cut-off frequency of the instrument response. For the joint inversion to function correctly, the depth resolution and sensitivity of the HVSR and DC data sets must be overlapping to some extent. When this is not the case, problems with convergence and partial over-fitting may occur, which can be observed with the present geophone inversion results. An example of overfitted data N0123 is displayed in Figure 4a,b, and an example of overconfident inversion results despite a poor data misfit, N1115, can be appreciated in Figure 4c,d. The data of N0123 are overfitted because the responses of the inversion results match near perfectly the mean observed data, but the confidence in the inversion results does not reflect the confidence in the observed data. While it is possible that the found model is indeed correct, it is more likely that some of the found features, such as the low velocity zones, are not required for a fit within data confidence and, thus, should be regarded as over-fitting artefacts. On the other hand, the inversion results of N1115 suggest a high confidence in the obtained model even though the data misfit is rather poor at least for the HVSR data. This means the convergence of the joint inversion failed consistently and only one of the two data sets, HVSR or DC, could be fitted well (for N1115 it is the DC data), and the data sets appear to be incompatible. There are three possibilities: (1) erroneous HVSR data, (2) erroneous DC data or (3) good data but the joint inversion was misguided due to a lack of correlation between the data and, thus, converged to a local minimum. Both effects, overfitting and overconfidence, are dominant for the inversion of geophone data; therefore, it is very likely that they occur due to the limited bandwidth available for HVSR and DC data with the geophone nodes and an ensuing depth sensitivity gap. While it is technically not possible to obtain longer periods of HVSR data from geophones, as discussed before, it may be possible to mitigate this problem by employing DC data of shorter periods, which were not available for this study (at the time of inquiry) but generally can be obtained if needed.

### 4.4. $V_{S}$ 3D Model and Comparison with Local-Earthquake Tomography

The smoothed and interpolated $V_{S}$ model correlates well in between stations in the sense that horizontal layered structures extend laterally over several sites, and several zones of low and high velocities can be appreciated. In general, the northern part is lower in $V_{S}$ than the southern part with some regional anomalies that align to some extent with features found in the geologic map. In particular, the north eastern part of the zone clearly indicates a basin with low $V_{S}$. Most interestingly, the southern contact of the basin with neighbouring formations shows a sharp step of the $V_{S}=2\;{\mathrm{kms}}^{-1}$ surface (see Figure 8). This could indicate the presence of a large fault and coincides approximately with a major fault, the North Pyrenean Frontal Thrust (see Figure 1 and Figure 2).

The independent $V_{S}$ model obtained from local earthquake tomography [26,27] and the presented results agree qualitatively well with the presence of lower velocity zones (see Figure 9). Note that LE tomography may overestimate the $V_{S}$ values at the nearer surface, i.e., above 1km, which can be observed readily by the near indiscernible changes in $V_{S}$ that are not likely to be so constant (even with the present cell size of 400m). At this depth, LE tomography should be rather observed as the upper limit of the model’s $V_{S}$ because, for the very shallow layers, the only cells that are sampled with LET are those immediately below a station. In addition, those cells are sampled only by upgoing parallel rays, thus without ray crossing vertical smearing limits the resolution of LET at shallow depth. The much finer variations obtained from the joint inversion are more plausible and are consistently below the values of the LE tomography results. In fact, it might be advantageous to utilise the joint inversion results as initial (or fixed) near surface model for a refined LE tomography inversion in order to relieve this type of inversion from the need to find a near surface model without available data sensitivity to the model parameters. This could potentially improve inversion convergence, data misfit and model resolution. Another possible strategy could be to include HVSR and DC Joint inversion in the tomography inversion, which could result in improved confidence in the model results as shown above for the inclusion of HVSR to the inversion of DC data.

### 4.5. Density Model

Large parts of the data set are relatively insensitive to density model variations as discussed above for the 1D joint inversion results. Nevertheless, at some depths and for some sites, density variation was necessary to achieve the best data fit. The low sensitivity of the density in our inversion prevents detailed interpretation of the results as low confidence in large parts of the results does not permit model slicing with sufficient confidence. We avoid this misrepresentation of confidence by only interpreting the mean density along depth columns of the model volume. The mean density is weighted by inverse depth to represent a decreasing effect of deep densities of equal size towards (hypothetical) gravity measurements and decreasing confidence with depth.

Figure 10 illustrates the mean density across the model between the depth of 1km and sea level. Zones of $2\;{\mathrm{tm}}^{-3}$ indicate that the data were largely insensitive to density variation in these zones in the model and may or may not be correct, but higher or lower density must have been required in the inversion to fit the data. Most high density zones of the model are consistently north of the the NPFT, hence, outside the Mauléon basin (see Figure 1) and might be related to Devonian materials below Miocene units. Notably, the western anomaly is placed exactly where there is an outcrop of jurassic/triassic formations. It appears that the density derived from HVSR and DC joint inversion may be reliable as long as the data are sensitive to it, which may not always be the case. However, it is possible to determine whether or not data sensitivity allows density interpretation, and whether its inclusion in the inversion in general does not show clear adverse effects for the recovery of $V_{S}$. Nevertheless, it might be a worthwhile consideration to combine this joint inversion with a joint gravity inversion in order to cross-validate the density model with all available data and potentially improve not only density but, although to a lesser extent, $V_{S}$ results too.

## 5. Conclusions

Large scale deployments for passive seismic data acquisition are common for various applications but data interpretation focus is typically on 3D imaging techniques such as local earthquake tomography and ambient noise tomography. Acquisition contains intrinsically HVSR measurements which are, however, often discarded due to the methods’ assumed limited depth sensitivity. In this work, we studied the utility and performance of HVSR in a large-N experiment with various instrument types.

We found, by performing joint inversions of HVSR and Rayleigh phase velocity DC data, that broadband and short period instruments performed better than geophone nodes due to a gap in sensitivity of the latter. It is most important that both data types overlap depth sensitivity, which will depend on the instrument inherent filter cut-off frequency for HVSR data and the availability of shortest periods for DC data. In our experiment setup, the geophones’ cut-off frequency was too high compared to the available period range for DC measurements. We observed that geophone data inversion resulted in overfitting of data and overconfidence in estimated model parameters, and we argued that these are symptoms for sensitivity gaps between the data types.

We conclude that HVSR data, in a joint inversion, can be used to compensate to some extent for the absence of short period DC data in order to improve resolution of shallower layers and/or avoid estimating short period DC, which are harder to obtain than HVSR. By including HVSR to DC inversions, we achieved confidence improvements of two to three times for the shallow layers that are, otherwise, not covered by the DC data. Furthermore, HVSR/DC joint inversion may be useful for generating initial models for 3D tomographic inversions in large scale deployments. Lastly, sensitivity to density of HVSR/DC joint inversion is situational and should be treated with care. While some subsurface structures may be sensitive, others are clearly not. Inclusion of gravity inversion to HVSR/DC joint inversion may be possible and prove useful.

## Figures and Tables

**Figure 1 sensors-21-05946-f001:**
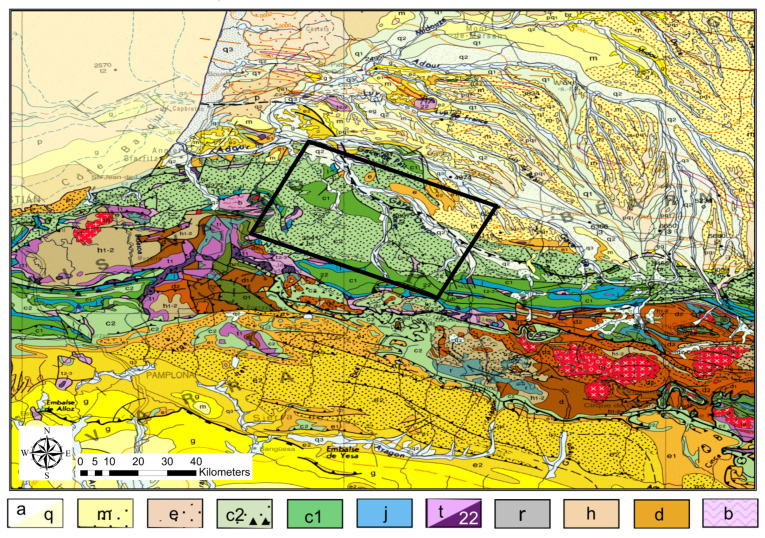
Regional geology depicting the predominantly cretaceous formations in the main area of the Maupasacq experiment. a: Quaternary alluvial deposits; q: Plio-Quaternary colluvium; m: Miocene; e: Eocene; c2: Upper Cretaceous; c1: Early Cretaceous; j: Jurassic; t: Triassic; 22: Ophites; r: Permian; h: Carboniferous; d: Devonian; b: Cambrian–Ordovician. Adapted from [21].

**Figure 2 sensors-21-05946-f002:**
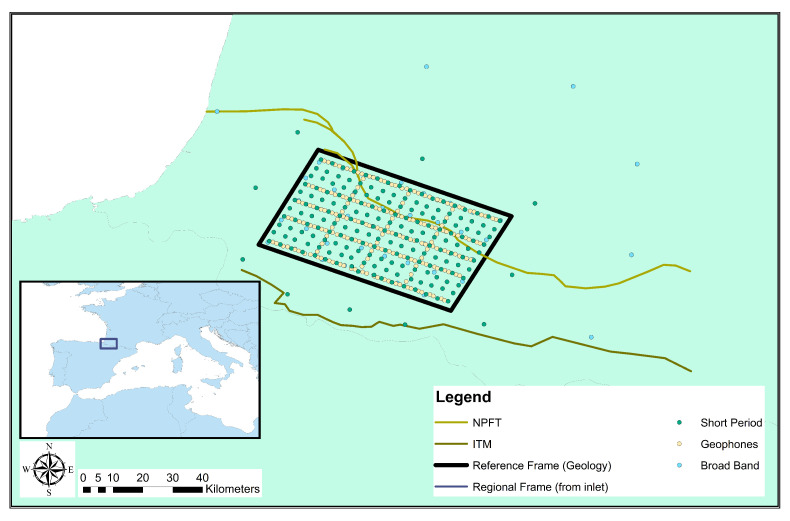
Placement and site layout of the Maupasacq experiment. Highlighted are site locations for the three employed sensor types and two regional fault systems: the North Pyrenean Frontal Thrust (NPFT) and the Igountze–Mendibelza Thrust (IMT).

**Figure 3 sensors-21-05946-f003:**
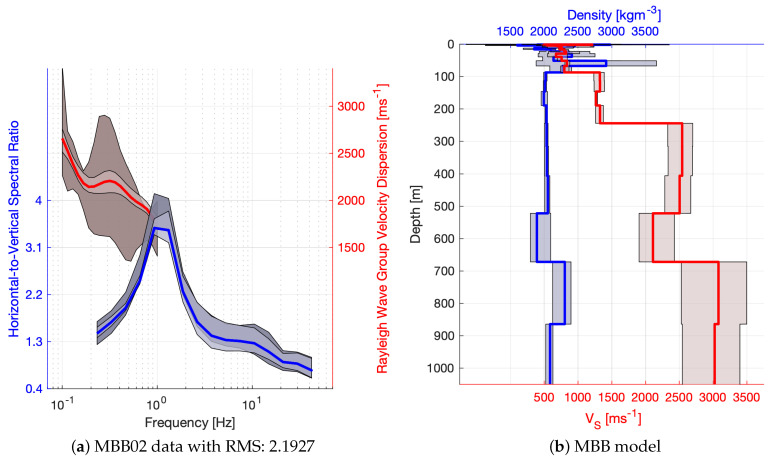
Examples of HVSR (blue) and DC (red) data with modelled responses (**left panels**) are displayed next to their respective 1D data inversion results (density blue and VS red; **right panels**). Light shades represent the confidence in the inversion results (model and data), while dark shades correspond to confidence in the measured data. Solid lines represent the weighted mean of the inversion results.

**Figure 4 sensors-21-05946-f004:**
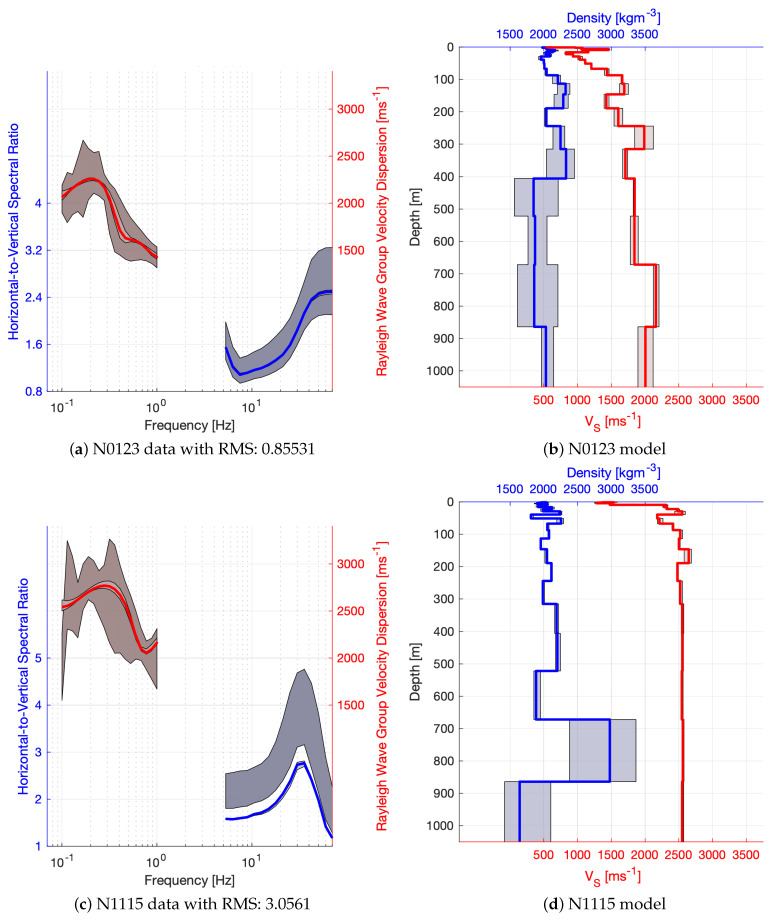
Examples of HVSR (blue) and DC (red) data with modelled responses (**left panels**) are displayed next to their respective 1D data inversion results (density blue and VS red; **right panels**). Light shades represent the confidence in the inversion results (model and data), while dark shades correspond to confidence in the measured data. Solid lines represent the weighted mean of the inversion results.

**Figure 5 sensors-21-05946-f005:**
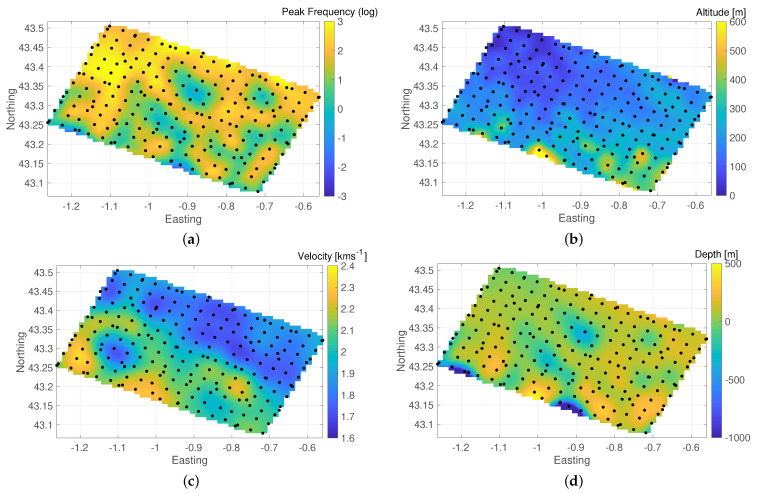
Maps associated to the peak frequency for the Maupasacq data set. Coordinates in degrees for latitude and longitude. (**a**) Smoothed peak frequency. (**b**) Altitude. (**c**) $V_{S}$ proxy. (**d**) Associated depth.

**Figure 6 sensors-21-05946-f006:**
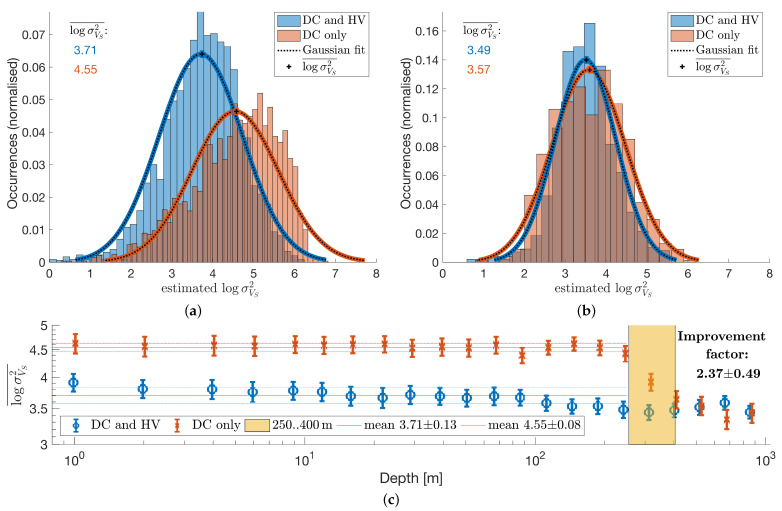
Final $V_{S}$ models’ variances are compared for DC inversions and joint inversions of DC and HVSR. (**a**,**b**) Depicted are distributions of variance estimates from short period data inversions for two distinct layer depth ranges and (**c**) the mean variance with 95% confidence for each layer. (**a**) Estimated $V_{S}$ variance (0to300m). (**b**) Estimated $V_{S}$ variance (300to1000m). (**c**) Estimated variances for each layer averaged over all sites.

**Figure 7 sensors-21-05946-f007:**
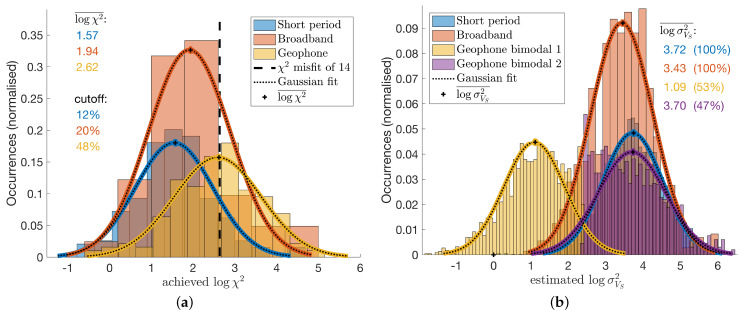
Final HVSR and DC joint inversion model statistics are compared for different instrument types. (**a**) Short period seismometers provide most consistent HVSR converging to lowest $\chi^{2}$ misfit. Broadbands are close second but half of the geophone data inversions did not reach an adequate misfit for a reliable interpretation. (**b**) Short period seismometers and broadbands provide comparable confidence, and geophone inversion’s confidence estimates yield a bi-modal distribution.

**Figure 8 sensors-21-05946-f008:**
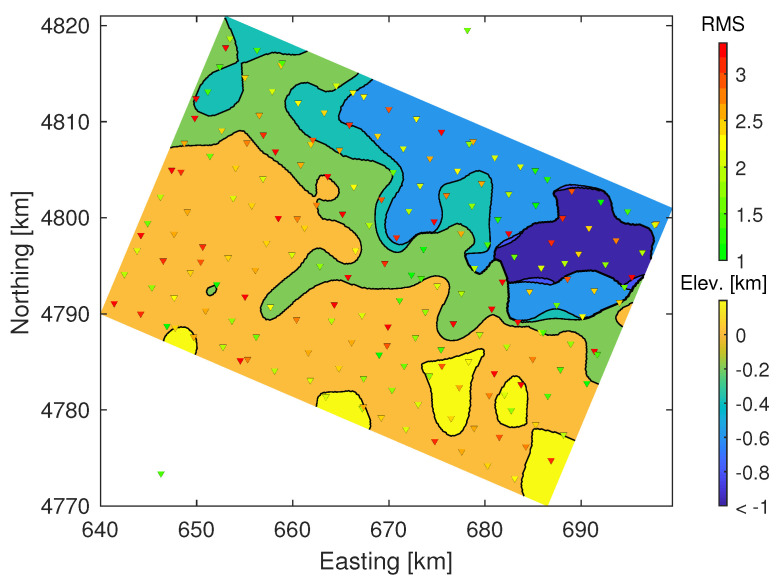
Depth of $V_{S}=2\;{\mathrm{kms}}^{-1}$ surface from 3D model projected on regional coordinates. Stations are marked as triangles with each face color corresponding to the station’s data achieved RMS misfit in the 1D joint inversion.

**Figure 9 sensors-21-05946-f009:**
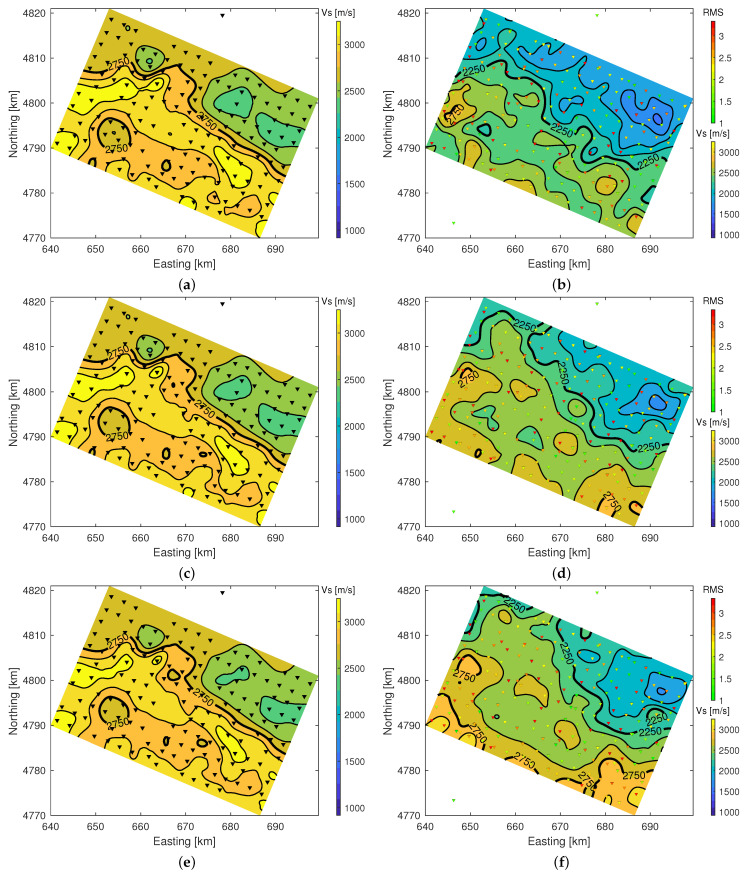
Contour plots illustrate $V_{S}$ results for LE tomography (**left**) and joint inversion of HVSR and DC (**right**). HVSR and DC joint inversion data fits (RMS) are displayed as marker colour for the triangular site location symbols. All maps projected on regional coordinates. (**a**) LET @200m depth. (**b**) HVSR/DC @200m depth. (**c**) LET @600m depth. (**d**) HVSR/DC @600m depth. (**e**) LET @1000m depth. (**f**) HVSR/DC @1000m depth.

**Figure 10 sensors-21-05946-f010:**
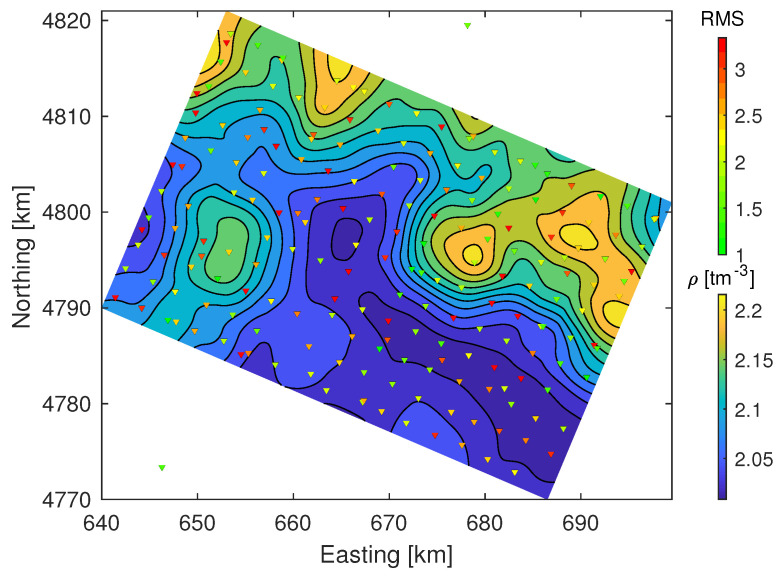
Mean density between depth of 1km and sea level projected on regional coordinates.

**Table 1 sensors-21-05946-t001:** One-dimensional model layer thicknesses and top depths in meters. Note that the lowest depth considered for the 3D model is 1000 m.

Top Depth	Layer Thicknesses			Top Depth
Initial Model	Step 1	Step 2	Steps 3–5	Final Model
0	1	1	1	0
1	8	3	1	1
			2	2
		5	2	4
			3	6
9	20	7	3	9
			4	12
		13	6	16
			7	22
29	58	22	10	29
			12	39
		36	16	51
			20	67
87	157	59	26	87
			33	113
		98	43	146
			55	189
244	428	162	71	244
			91	315
		266	116	406
			150	522
672	1163	439	192	672
			247	864
		724	317	1111
			407	1428
1835	3165	1194	523	1835
			671	2358
		1971	863	3029
			1108	3892
5000	*∞*	*∞*	*∞*	5000

**Table 2 sensors-21-05946-t002:** One-dimensional inversion strategy and model parameter bounds.

Step	# Layers	$V_{P}$	$V_{S}$	Density
1	9	0.4 to $7\;\mathrm{km}/s^{-1}$ (free)	0.2 to $4\;\mathrm{km}/s^{-1}$ (increasing)	$2\;t/m^{-3}$ (constant)
2	16			
3	30			
4	30		0.2 to $4\;\mathrm{km}/s^{-1}$ (free)	
5	30			1 to $3\;t/m^{-3}$ (free)

## Data Availability

The waveform data used in this study will be made publicly available after an embargo period. Additionally, HVSR curves and derived models are available from the corresponding author upon request.

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
