# Peer review of "On the Utility of Horizontal-to-Vertical Spectral Ratios of Ambient Noise in Joint Inversion with Rayleigh Wave Dispersion Curves for the Large-N Maupasacq Experiment"

_sensors, 2021, doi:10.3390/s21175946_

Round 1
Reviewer 1 Report
This paper presents a case study of joint inversion of HVSR and dispersion curves for the Vs structures. Generally, it is well-written and indicates the advantages of joint inversion. I only have some minor comments on the data processing. How did the authors extract the DCs and HVSR from raw data? What method did the authors use?How did they calculate the uncertainties? Did the author consider the influence of different modes of surface waves on the estimation of HVSR curves? Because the interference of different Rayleigh-wave modes may introduce
errors in the calculation of HVSR for a specific mode (Mi et al., 2019).
Reviewer 2 Report
Review of manuscript "On the Utility of Horizontal-to-Vertical Spectral Ratios of Ambient Noise in Joint Inversion with Rayleigh Wave Dispersion Curves for the Large-N Maupasacq Experiment" by Maik Neukirch, Antonio García-Jerez, Antonio Villaseñor, Francisco Luzón, Jacques Brives and Laurent Stehly, submitted to Sensors Journal, Special Issue "Data Acquisition and Analysis of Seismic Noise".
This is an interesting work on the usefulness and advantages of employing HVSR of ambient noise for the joint inversion of surface waves dispersion curves, in this case Rayleigh waves.
Manuscript is well structured, well organized and written in good English language. The methods employed for the analysis are quite adequate and the results are thoroughly analyzed. Conclusions are consistent with the objectives of the study. I have no important, or major, concerns on the data analysis procedures, inversion methodologies or results. However, I have some minor concerns that should be addressed:
Specific comments:
1- Even when abbreviations are shown and explained in lines 379-382, please include the abbreviation of the term Dispersion Curves (DC) in line 2 of abstract, in the same way it is included in line 58 of the text.
- The contour shape of the Maupasacq experiment shown in Figures 1, 2 and 5 appear different (diamond shaped) from those in Figures 8, 9 and 10 (rectangular). Maybe this is due to the geographical projection used, or a possible perspective of map view. Please clarify this, as it is not evident why authors are presenting maps and results in these two ways.
- In lines 129-130 of section 3, authors mention that "...we consider DC data and LET inversion results as given by other studies, presented elsewhere". This sentence is somehow confusing. Authors should include here, the main references used for this purpose in this work.
- A similar case occurs in lines 140-141: "...however, this method and details on obtained Rayleigh wave group velocity maps will be discussed elsewhere". Again, the last part of this sentence is confusing. Please clarify and include references.
- Lines 156-157. Please include in text here that the "Xi2>14" assumed to discard stations is explained in sections 4.2 and 4.3.
- Please rewrite text in lines 157-159; commas are badly placed; In this same sentence: Are Depths relative to the mean altitude of the seismic network, or to the mean sea level? This is not clear.
- Lines 280-281. Please explain why the Dispersion Curves (DC) data of shorter periods were not available for this study? It appears that this was only due to the availability of the data, and not because of other possible reasons (low signal-to-noise ratio in high frequencies, sampling rates, or maybe other possible inherent problems when obtaining or calculating the DC).
- Line 297. What does the term "mathrmm" means? Is this from a LaTex Symbol - meters? Please check.
- Please check the reference list. Some references are incomplete.
Minor general comments
- I couldn't find along manuscript the description of the instrumental response corrections performed by authors before employing the different types of seismic data for the analysis. I suggest authors to include a brief description of the corrections performed in reference [17; Polychronopoulou, et al. 2018], on each type of seismic data, given the three different types of instruments used in this study.
- Lines 332-335 on the density model obtained derived from the joint inversion. As authors comment, it would be necessary to include here a joint inversion with gravity data, as this result might become speculative. In this specific case, it could be useful to include a comparison with some previous general Bouguer anomaly study for the region.
Given the above comments I recommend a MINOR REVISION before the manuscript is published in Sensors, Special Issue "Data Acquisition and Analysis of Seismic Noise".
